# Partially methylated domains are hypervariable in breast cancer and fuel widespread CpG island hypermethylation

Arie B. Brinkman [1], Serena Nik-Zainal[2,3], Femke Simmer[1,17], F. Germán Rodríguez-González[4], Marcel Smid[4], Ludmil B. Alexandrov[2,5,6], Adam Butler[2], Sancha Martin [2], Helen Davies[2], Dominik Glodzik[2], Xueqing Zou[2], Manasa Ramakrishna[2], Johan Staaf [7], Markus Ringnér [7], Anieta Sieuwerts [4], Anthony Ferrari[8], Sandro Morganella[9], Thomas Fleischer [10], Vessela Kristensen[10,11,12], Marta Gut[13], Marc J. van de Vijver[14], Anne-Lise Børresen-Dale[10,11], Andrea L. Richardson[15,16], Gilles Thomas[8], Ivo G. Gut[13], John W.M. Martens [4], John A. Foekens[4], Michael R. Stratton [2] & Hendrik G. Stunnenberg [1]

Global loss of DNA methylation and CpG island (CGI) hypermethylation are key epigenomic aberrations in cancer. Global loss manifests itself in partially methylated domains (PMDs) which extend up to megabases. However, the distribution of PMDs within and between tumor types, and their effects on key functional genomic elements including CGIs are poorly defined. We comprehensively show that loss of methylation in PMDs occurs in a large fraction of the genome and represents the prime source of DNA methylation variation. PMDs are hypervariable in methylation level, size and distribution, and display elevated mutation rates. They impose intermediate DNA methylation levels incognizant of functional genomic elements including CGIs, underpinning a CGI methylator phenotype (CIMP). Repression effects on tumor suppressor genes are negligible as they are generally excluded from PMDs. The genomic distribution of PMDs reports tissue-of-origin and may represent tissue-specific silent regions which tolerate instability at the epigenetic, transcriptomic and genetic level.

---

[1] Department of Molecular Biology, Faculty of Science, Radboud Institute for Molecular Life Sciences, Radboud University, PO Box 9101, Nijmegen 6500 HB, The Netherlands. [2] Wellcome Trust Sanger Institute, Hinxton, Cambridge CB10 1SA, UK. [3] Academic Department of Medical Genetics, University of Cambridge, Cambridge CB2 0QQ, UK. [4] Erasmus MC Cancer Institute and Cancer Genomics Netherlands, Department of Medical Oncology, Erasmus University Medical Center, Rotterdam 3015 GD, The Netherlands. [5] Theoretical Biology and Biophysics (T-6), Los Alamos National Laboratory, Los Alamos, NM 87545, USA. [6] Center for Nonlinear Studies, Los Alamos National Laboratory, Los Alamos, NM 87545, USA. [7] Division of Oncology and Pathology, Department of Clinical Sciences Lund, Lund University, Lund SE-223 81, Sweden. [8] Synergie Lyon Cancer, Centre Léon Bérard, 28 rue Laënnec, Lyon Cedex 08, France. [9] European Molecular Biology Laboratory, European Bioinformatics Institute, Wellcome Trust Genome Campus, Hinxton, Cambridgeshire CB10 1SD, UK. [10] Department of Genetics, Institute for Cancer Research, Oslo University Hospital, The Norwegian Radium Hospital, Oslo 0310, Norway. [11] K.G. Jebsen Centre for Breast Cancer Research, Institute for Clinical Medicine, University of Oslo, Oslo 0316, Norway. [12] Department of Clinical Molecular Biology and Laboratory Science (EpiGen), Division of Medicine, Akershus University Hospital, Lørenskog 1478, Norway. [13] Centro Nacional de Análisis Genómico (CNAG), Parc Científic de Barcelona, Barcelona 08028, Spain. [14] Department of Pathology, Academic Medical Center, Meibergdreef 9, Amsterdam AZ 1105, The Netherlands. [15] Department of Pathology, Brigham and Women's Hospital, Boston, MA 02115, USA. [16] Dana-Farber Cancer Institute, Boston, MA 02215, USA. [17]Present address: Department of Pathology, Radboud University Nijmegen Medical Centre, P.O. Box 9101, Nijmegen 6500 HB, The Netherlands. Correspondence and requests for materials should be addressed to A.B.B. (email: a.brinkman@science.ru.nl) or to H.G.S. (email: h.stunnenberg@ncmls.ru.nl)

Global loss of methylation was among the earliest recognized epigenetic alterations of cancer cells[1]. It is now known to occur in large genomic blocks that partially lose their default hypermethylated state, termed partially methylated domains (PMDs)[2–6]. PMDs have been described for a variety of cancer types and appear to represent repressive chromatin domains that are associated with nuclear lamina interactions, late replication, and low transcription. PMDs are not exclusive to cancer cells and have also been detected in normal tissues[2,7–12], but are less pronounced in pluripotent cells and brain tissue[12–14]. PMDs can comprise up to half of the genome[3,4,12], and it has been suggested that PMDs in different tissues are largely identical[3,12]. PMDs have been shown to harbor focal sites of hypermethylation that largely overlap with CGIs[3]. Questions remain as to what instigates such focal hypermethylation, whether loss of methylation inside PMDs is linked to repression of cancer-relevant genes and whether the genomic distribution of PMDs is invariant throughout primary tumors of the same type, perhaps determined by tissue-of-origin. In breast cancer, PMDs have been detected in two cultured cancer cell lines[5], but their extent and variation in primary tumors is hitherto unknown. A major limitation of most DNA methylation studies is that only a small subset of CpGs are interrogated. This prevents accurate determination of the extent and location of PMDs. Few samples of a certain tissue/tumor have typically been analyzed using whole-genome bisulfite sequencing (WGBS). Thus, observations cannot be extrapolated to individual cancer types. Here, we analyzed DNA methylation profiles of 30 primary breast tumors at high resolution through WGBSs. This allowed us to delineate breast cancer PMD characteristics in detail. We show that PMDs define breast cancer methylomes and are linked to other key epigenetic aberrations such as CGI hypermethylation.

## Results

**Primary breast tumors show variable loss of DNA methylation.** To study breast cancer epigenomes we performed WGBS in 30 primary breast tumors, encompassing ~95% of annotated CpGs (Supplementary Fig. 1A, Supplementary Data 1). For 25/30 of these tumors we previously analyzed their full genomes[15,16] and transcriptomes[17], respectively. Of the 30 tumors, 25 and 5 are ER-positive and ER-negative, respectively (Supplementary Fig. 1B, Supplementary Data 2).

To globally inspect aberrations in DNA methylation patterns we generated genome-wide and chromosome-wide methylome maps by displaying mean methylation in consecutive tiles of 10 kb (see Methods section). These maps revealed extensive inter-tumor variation at genome-wide scale (Fig. 1a). At chromosome level, we observed stably hypermethylated regions next to regions that were hypomethylated to various extents and across tumors (Fig. 1b). Chromosomes 1 and X were exceptionally prone to methylation loss, the latter of which may be related to epigenetic aberrations of the inactive X-chromosome in breast cancer observed by others[18]. At megabase scale (Fig. 1c) DNA methylation profiles showed that the widespread loss of methylation occurred in block-like structures previously defined as PMDs[2]. Across primary breast tumor samples, DNA methylation levels and genomic sizes of PMDs differ extensively between tumors and PMDs do appear as separate units in some tumors and as merged or extended in others, underscoring the high variation with which methylation loss occurs. Despite this variation, however, we observed common PMD boundaries as well.

Given the variation between tumors, we asked whether the patterns of methylation loss were associated with distribution of copy-number variations (CNVs) throughout the genome.

We found no evidence for such association (Pearson $R = 0.17$), although we noticed that chromosomes with the most pronounced loss of methylation (chr1, chrX, and chr8-p) frequently contained amplifications (Supplementary Fig. 1C). Next, we asked whether loss of methylation was associated with aberrant expression of genes involved in writing, erasing, or reading the 5-methylcytosine modification. However, we found no such correlation (Supplementary Fig. 1D). Finally, we assessed whether mean PMD methylation was associated with the fraction of aberrant cells within the sample (ASCAT[19]). However, no such correlation was evident (Pearson $R = -0.03$, Supplementary Fig. 1E).

To provide a reference for the observed patterns of methylation loss we compared WGBS profiles of primary breast tumors to that of 72 normal tissues (WGBS profiles from Roadmap Epigenomics Project and in ref. [10], Supplementary Fig. 2A,B). In sharp contrast to breast cancer, most normal tissues were almost fully hypermethylated (except for pancreas and skin), with heart, thymus, embryonic stem cell(-derived), induced pluripotent stem cells and brain having the highest levels of methylation. Importantly, inter-tissue variation was much lower as compared to breast tumors ($p < 2.2e-16$, MWU-test on standard deviations). The variation observed among breast tumors was also present when we reproduced Fig. 1a–c using only solo-WCGW CpGs (CpGs flanked by an A or T on both sides), which were recently shown to be more prone to PMD hypomethylation[12] (Supplementary Fig. 3). Thus, breast tumors show widespread loss of DNA methylation in PMDs, and the extent and patterns appear to be hypervariable between tumor samples. In line with this, principal component analysis confirmed that methylation inside PMDs is the primary source of variation across full-genome breast cancer DNA methylation profiles (Fig. 1d): the first principal component (PC1) is strongly associated with mean PMD methylation ($p = 3.2e-07$[20], see Methods section). The second-largest source of variation, PC2, is associated with ER status ($p = 2.7e-07$, Fig. 1d, Supplementary Fig. 4A, see Methods section) and to a lesser extent with intrinsic AIMS subtypes (Absolute assignment of breast cancer Intrinsic Molecular Subtypes, $p = 4.2e-04$, see Methods section, Supplementary Fig. 4A)[21,22], although the latter is likely confounded with ER status. Successive PCs were not significantly associated with any clinicopathological feature. It should be noted that with 30 tumors only very strong associations can achieve statistical significance. Taken together, breast tumor whole-genome DNA methylation profiles reveal global loss of methylation in features known as PMDs, the extent of which is hypervariable across tumors and represent the major source of variation between tumors.

**Distribution and characteristics of breast cancer PMDs.** We set out to further characterize breast cancer PMDs and their variation (see Methods: data availability). The genome fraction covered by PMDs varies greatly across our WGBS cohort of 30 tumors, ranging between 10 and 50% across tumors, covering 32% of the genome on average (Fig. 2a). We define PMD frequency as the number of tumors in which a PMD is detected. A PMD frequency of 30 (PMDs common to all 30 cases) occurs in only a very small fraction of the genome (2%), while a PMD frequency of 1 (representing the union of all PMDs from 30 cases) involves 70.2% of the genome (Fig. 2b). Similar results were obtained with PMDs called on only solo-WCGW CpGs[12] (Supplementary Fig. 4B,C), and comparison of these solo-CpG PMDs with "all-CpG" PMDs revealed high overlap (92%) between their individual unions (Supplementary Fig. 4D). We further compared our PMD calling with aggregate PMD calling based on cross-

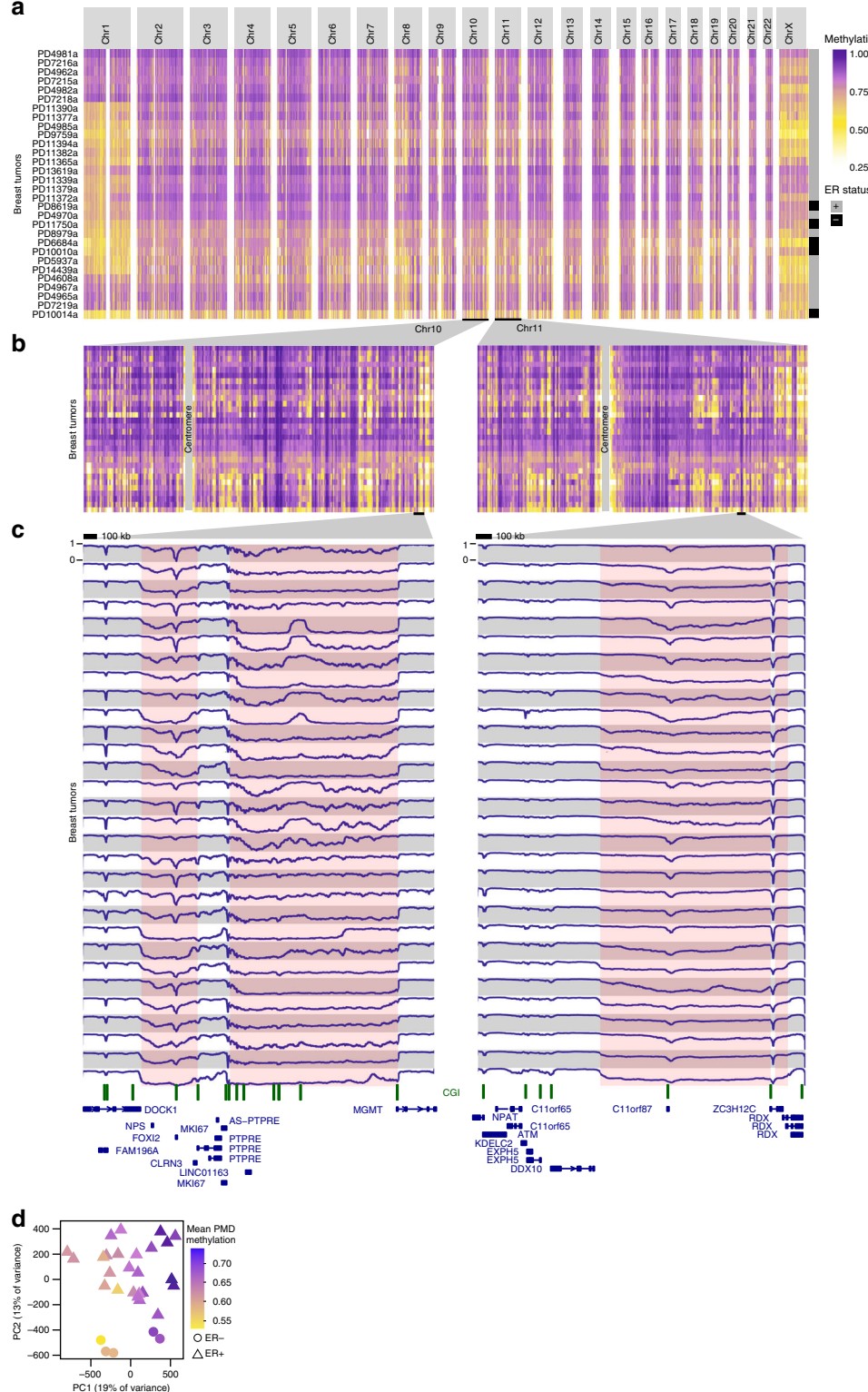

**Fig. 1** Visualization of inter-tumor variation at genome-wide scale. **a** Genome-wide and **b** chromosome-wide maps of WGBS DNA methylation profiles from 30 breast tumor samples. Mean methylation is displayed in consecutive tiles of 10 kb (see Methods section). Ordering of tumor samples is according clustering of the tiled profiles. **c** WGBS DNA methylation visualization at megabase-scale. Pink coloring indicates common methylation loss (PMDs), although tumor-specific PMD borders vary. A scale bar (100 kb) is shown at the top of each panel. CpG islands are indicated in green. **d** Principal component analysis of WGBS DNA methylation profiles (see Methods section). Each tumor sample is represented with its estrogen-receptor (ER) status (point shape) and mean PMD methylation (point color)

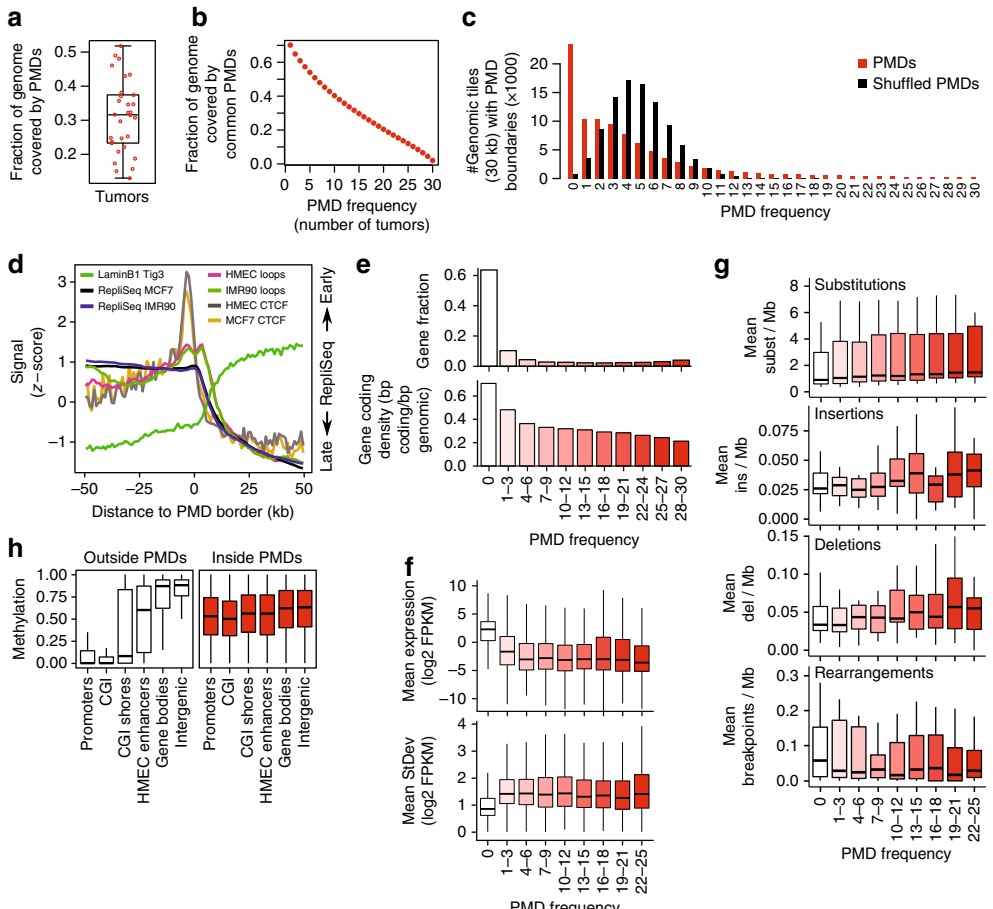

**Fig. 2** Characterization of breast cancer PMDs. **a** Fraction of the genome covered by PMDs. Each dot represents one tumor sample, the boxplot summarizes this distribution. **b** Fraction of the genome covered by PMDs that are common between breast tumors. PMD frequency: the number of tumors in which a genomic region or gene is a PMD. **c** Breast cancer PMDs are not distributed randomly over the genome. The genome was dissected into 30-kb tiles, PMD frequency (number of boundaries) was calculated for each tile. The same analysis was done after shuffling the PMDs of each tumor sample. **d** Average profiles of LaminB[23], repliSeq (DNA replication timing, ENCODE), 3D chromatin interaction loops (HiC[27], and CTCF (ENCODE) over PMD borders. If available, data from the breast cancer cell line (MCF7) and mammary epithelial cells (HMEC) was used, otherwise data from fibroblasts (IMR90, Tig3) was used. **e** Gene distribution inside PMDs (top, as a fraction of all annotated genes; bottom, as gene coding density). **f** Gene expression inside PMDs. Gene expression (top) and standard deviation (bottom) for the 25 overlapping cases of our WGBS and the transcriptome cohorts[17] was plotted as a function of PMD frequency. **g** Somatic mutations inside PMDs. Substitutions, insertions, deletions, and rearrangements were calculated for the 25 overlapping cases of our WGBS and the breast tumor full genomes cohorts[15], and plotted as a function of PMD frequency. **h** Distribution of DNA methylation over functional genomic elements, inside and outside PMDs. CpGs were classified according PMD status and genomic elements, and the distribution of DNA methylation within each element was plotted. All boxplots in this figure represent the median and 25th and 75th percentiles, whiskers 1.5 times the interquartile range, outliers are not shown

sample standard deviation (s.d.) of methylation in 100-kb genomic bins[12]. This method segments the genome according to common PMDs across multiple samples, and we found that our PMDs are all contained within this aggregate PMD track (Supplementary Fig. 4E,F).

Given the inter-tumor variation of PMDs we tested to which extent PMD distribution is random by counting PMD borders in 30-kb genomic tiles (Fig. 2c). Randomly shuffled PMDs yield a normal distribution centered at a PMD frequency of four. In contrast, observed PMDs show a skewed distribution: the mode was for a PMD frequency of 0 suggesting that many tiles (23,492, 25%) do not coincide with any PMD borders. The majority of tiles (62%) had a low PMD border frequency (1–10). The tail represents low numbers of tiles with up to maximal PMD frequency of 30. We conclude that PMD distribution is not random: part of the genome appears not to tolerate PMDs while PMDs occur in a large fraction of the genome with varying frequencies.

PMDs have been shown to coincide with lamin-associated domains (LADs)[3,4]: large repressive domains that preferentially locate to the nuclear periphery[23]. LADs are characterized by low gene density and late replication[23,24]. Accordingly we found that PMDs show reduced gene densities (Fig. 2e), have high LaminB1 signals (associated with LADs[23], Fig. 2d), are late replicating (ENCODE data, Fig. 2d) and have a low frequency of (Hi-C) 3D loops[25], an indicator of lower levels of transcription. Finally, we observed a local increase in binding of the transcription factor CTCF at the borders of PMDs (Fig. 2d) as shown in previous reports[3,23,26–28].

We previously analyzed the full transcriptomes (RNA-seq) in a breast cancer cohort of 266 cases[17] from which our WGBS cohort is a subset. We determined the mean expression of genes as a function of PMD frequency in the overlapping subset of 25 tumors. Genes inside PMDs are expressed at consistently lower levels than genes outside of PMDs (Fig. 2f, p < 2.2e−16, t-test), with a tendency towards lower expression in highly-frequent

PMDs ($p < 2.2e{-}16$, linear regression). Given the variable nature of DNA methylation patterns of PMDs, we also determined the variation (s.d.) in gene expression as a function of PMD frequency and found higher variation for genes inside PMDs (Fig. 2f, $p < 2.2e{-}16$, MWU-test). When extending this analysis to the full set of 266 cases from the transcriptome cohort we observed the same (Supplementary Fig. 5A, $p < 2.2e{-}16$, $t$-test for expression; $p < 2.2e{-}16$, MWU-test for variation). Given the observed variability of DNA methylation and gene expression inside PMDs, we asked whether genetic stability, i.e., the number of somatic mutations, was also altered within PMDs. In the 25 overlapping cases between our WGBS cohort and the WGS cohort[15], substitutions, insertions, and deletions occur more frequently within than outside PMDs ($p < 0.0005$ for each mutation type, logistic regression), with a (slight) increase in highly frequent PMDs ($p < 2.2e{-}16$ for substitutions, $p = 0.37$ for insertions, $p = 1.6e{-}05$ for deletions, logistic regression, Fig. 2g). In contrast, rearrangements are more abundant outside of PMDs ($p = 1.1e{-}09$, logistic regression), in keeping with the hypothesis that regions with higher transcriptional activity are more susceptible to translocations[29]. We extended this analysis to the full cohort of 560 WGS tumor samples[15], which confirmed these observations while showing much stronger effects in highly frequent PMDs ($p < 2.2e{-}16$ for all mutation types and rearrangements, logistic regression, Supplementary Fig. 5B). Taken together, breast cancer PMDs share key features of PMDs including low gene density, low gene expression, and colocalization with LADs, suggesting that they reside in the B (inactive) compartment of the genome[30]. Importantly, in addition to epigenomic instability, breast cancer PMDs also tolerate transcriptomic variability and genomic instability.

**CpG island methylation in breast cancer PMDs.** To determine how PMDs affect methylation of functional genomic elements we accordingly stratified all CpGs from all tumors and assessed the methylation distribution in these elements (Fig. 2h). We found that the normally observed near-binary methylation distribution is lost inside PMDs; the hypermethylated bulk of the genome and hypomethylated CGIs/promoters acquire intermediate levels of DNA methylation inside PMDs. DNA methylation deposition inside PMDs thus appears incognizant of genomic elements, resulting in intermediate methylation levels regardless of the genomic elements' functions. Among all elements, the effect of incognizant DNA methylation deposition is most prominent for CGIs as they undergo the largest change departing from a strictly hypomethylated state. This has been described also as focal hypermethylation inside PMDs[3].

We further focused on methylation levels of CGIs. When indiviual PMDs are regarded, CGIs inside of them lose their strictly hypomethylated state and become more methylated to a degree that varies between tumors (Fig. 3a). Across all tumors and all CGIs, this effect is extensive (Fig. 3b, c), affecting virtually all CGIs inside PMDs: on average 92% of CGIs lose their hypomethylated state and gain some level of methylation (Fig. 3b, left panel). Outside of PMDs only 25–30% of the CGIs is hypermethylated, although to a higher level (Fig. 3b, right panel). Thus, incognizant deposition of DNA methylation inside PMDs results in extensive hypermethylation of virtually all PMD-CGIs.

Concurrent hypermethylation of CGIs in cancer has been termed CIMP[31], and in breast cancer this phenomenon has been termed B-CIMP[32–34]. To determine whether CIMP is directly related to PMD variation we defined B-CIMP as the fraction of CGIs that are hypermethylated (>30% methylated), and determined its association with the fraction of CGIs inside PMDs. Regression analysis (see Methods) showed that this association is highly significant (Fig. 3f, $p = 2.1e{-}08$, $R^2 = 0.51$, $n = 30$). The fraction of hypermethylated CGIs is generally higher than the fraction of hypermethylated CGIs in PMDs, suggesting that CGI hypermethylation is not solely dependent on PMD occurrence. However, CGI methylation levels outside PMDs are far more stable than inside PMDs (Fig. 3e), which likely represents an invariably methylated set of CGIs (Supplementary Data 3).

We applied the same regression analysis to 14 other tumor types (TCGA[35], BLUEPRINT[36–38], Fig. 3g). Although sample sizes were small, we found significant CIMP-PMD associations for lung adenocarcinoma (LUAD), rectum adenocarcinoma (READ), uterine corpus endometrial carcinoma (UCEC), and bladder urothelial carcinoma (BLCA). We did not find significant associations for other tumor types (ALL, BL, ALL, CLL, FL, LUSC, lung, TPL, STAD, MCL, BLCA, see Fig. 4b for their abbreviations) and glioblastoma (GBM), even though for the latter G-CIMP has been previously described[39]. Taken together, we conclude that PMD occurrence is an important determinant for CIMP in breast cancer and a subset of other tumor types.

**PMD demethylation effects on gene expression.** To assess whether widespread hypermethylation of CGI-promoters within PMDs instigates gene repression we analyzed expression as a function of gene location inside or outside of PMDs. Overall, CGI-promoter genes showed a mild but significant down-regulation when inside PMDs ($p = 4.5e{-}12$, $t$-test), while strong downregulation was specifically restricted to low-frequency PMDs (Fig. 3h). For non-CGI-promoter genes this trend was very weak or absent (Supplementary Fig. 6A). As healthy controls were not included in transcriptome analysis of our cohort[17] we used gene expression (RNA-seq) profiles from breast tumors (769) and normal controls (88) from TCGA. Similar to our cohort (see Fig. 2f) we found that overall gene expression for the TCGA tumors is lower inside PMDs, with lowest expression for genes inside high-frequent PMDs (Fig. 3i, $p < 2.2e{-}16$, linear regression). However, the expression of genes in tumor PMDs is very similar to healthy control samples ($p = 0.807$, linear regression). To analyze this in more detail we selected normal/tumor matched pairs (i.e., from the same individuals, $n = 86$) and analyzed the fold change over the different PMD frequencies (Fig. 3j). As in our cohort, downregulation is restricted to genes with low PMD-frequency ($p < 2.2e{-}16$ for PMD frequency 1–3, linear regression). No obvious changes occur in high-frequency PMD genes, nor in non-CGI-promoter genes (Supplementary Fig. 6B). Taken together, widespread cancer-associated repression of all genes inside PMDs is limited: downregulation is restricted to low-frequency (i.e., the more variable) PMDs and affects only CGI-promoter genes, which undergo widespread hypermethylation inside PMDs.

Given the widely accepted model of hypermethylated promoter-CGIs causing repression of tumor suppressor genes (TSGs) we determined whether breast cancer PMDs overlap with these genes to instigate such repression. For non-TSGs as a reference we found that 64% (14,037) are located outside of PMDs (Fig. 3k), while 36% are located inside, (see also Fig. 2e). Strikingly, TSGs (Cancer Gene Census) overlap poorly with PMDs: most TSGs (218/254, 86%) are located outside of PMDs. Only 14% overlap with mostly low-frequency PMDs, implying exclusion of TSGs from PMDs ($p = 8.8e{-}16$, hypergeometric test). When we specifically focused on breast cancer-related TSGs (Cancer Gene Census), this exclusion was even stronger: practically all (27/28, 96%) breast cancer TSGs are located outside of PMDs ($p = 3.5e{-}06$, hypergeometric test). Similarly, from our previously identified set of genes containing breast cancer driver mutations[15]: 86/93 (92%) were located outside of PMDs ($p = 2.0e{-}11$, hypergeometric test). Alltogether, only 31

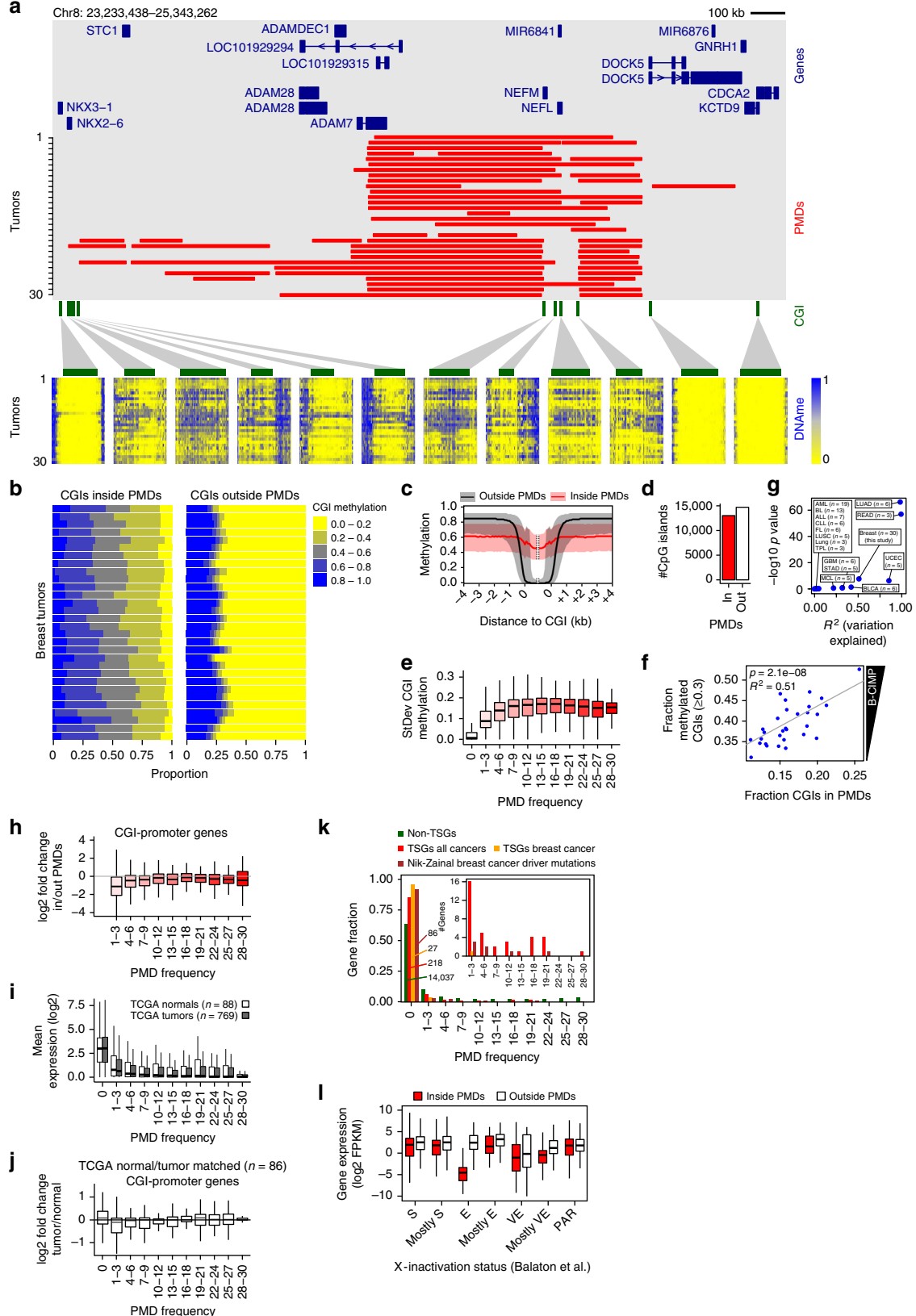

breast cancer-mutated genes were not excluded from PMDs. We assessed whether these genes are downregulated in tumors when inside PMDs. 24/31 (74%) genes were downregulated (Supplementary Fig. 7A, B), and an overall negative correlation between CGI-promoter methylation and expression was evident (Supplementary Fig. 7C). For 16 out of these 24 genes we confirmed that significant downregulation also takes place in cancer relative to normal in an independent breast cancer expression dataset (TCGA, see examples in Supplementary Fig. 7D). Among the downregulated genes in PMDs are *EGFR*

**Fig. 3** CpG island hypermethylation inside PMDs. **a** Representative 2.1-Mb genomic region. Red bars, PMDs for each tumor; below, CGI methylation per tumor (same ordering). Green bars, CGIs. **b** CGI methylation as the fraction of all CGIs (x-axis). Horizontal bars represent individual tumors. **c** Methylation over CGIs inside/outside PMDs, averaged over all 30 tumors. Black/red lines, median; gray/pink area, 1st and 3rd quartiles. **d** CGI counts inside and outside of breast cancer PMDs. "in", CGIs inside PMDs in at least one tumor sample. **e** Variation of CGI methylation (standard deviation) as a function of PMD frequency. **f** Regression analysis of B-CIMP (y-axis) as a function of the fraction CGIs inside PMDs (x-axis). B-CIMP: the genome-wide fraction of hypermethylated CGIs (>30% methylation). **g** Summary of regression analyses as in (F), for additional cancer types. n, the number of samples for each type. For abbreviations of cancer type names, see Fig. 4b. **h** Expression change of CGI-promoter genes inside vs. outside of PMDs, as a function of PMD frequency. **i** Gene expression as a function of PMD frequency in TCGA breast cancer data. PMD frequency was derived from our own methylation data. **j** Expression change of CGI-promoter genes as a function of PMD frequency in matched breast cancer tumor/normal pairs (TCGA). PMD frequency was derived from our own methylation data. **k** Tumor-suppressor genes (TSGs) are excluded from PMDs. PMD frequency was determined for each TSG and the resulting distribution was plotted. Main plot, relative distribution; inset, absolute gene count. "Non-TSGs", genes not annotated as TSGs; "TSGs all cancers", genes annotated as TSGs regardless of cancer type; "TSGs breast cancer", genes annotated as TSG in breast cancer; "Nik-Zainal breast cancer driver mutations", genes with driver mutations in breast cancer[15]. **l** Expression of X-linked genes when inside or outside PMDs. Genes were grouped according X-inactivation status (E, escape; S, subject to XCI; VE, variably escaping; PAR, pseudoautosomal region)[41]. All boxplots in this figure represent the median and 25th and 75th percentiles, whiskers 1.5 times the interquartile range, outliers are not shown

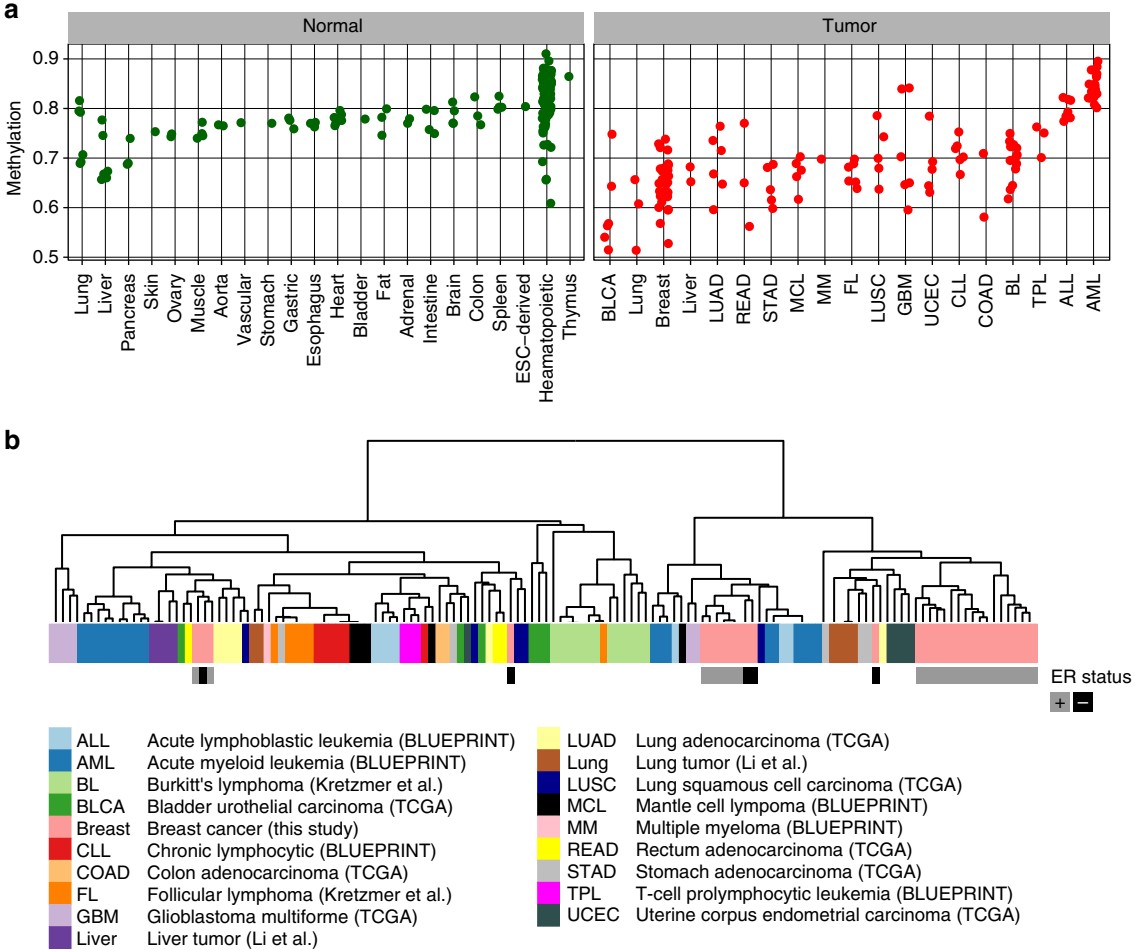

**Fig. 4** PMD methylation in normal tissues and tumors of various tissues. **a** Mean PMD methylation of normal tissues and tumors of various tissue types. Each dot represents one sample. **b** Hierarchical clustering of tumor samples based on genomic distribution of their PMDs. For breast tumors (this study) the ER status is indicated

(epidermal growth factor receptor) and *PDGFRA* (platelet-derived growth factor receptor α) that have tumor promoting mutations (Supplementary Fig. 7A–C). Paradoxically, both genes are significantly downregulated in our as well as the TCGA breast cancer dataset (Supplementary Fig. 7D). Taken together, despite the large number of hypermethylated CpG islands inside breast cancer PMDs (13,013 CGIs; 47%, Fig. 3d), these CGIs do not generally co-occur with TSGs and other breast cancer-relevant genes. Repression of these genes through classical promoter-

hypermethylation in PMDs does not occur at large scale, and is likely limited to a few genes.

We next identified genes that are downregulated when inside PMDs regardless of any documented TSG function or mutation in breast cancer. Four hundred genes were downregulated at least 2.5 log2-fold (Supplementary Data 4). Gene set enrichment analysis showed that these genes were involved in processes such as signaling and adhesion (Supplementary Fig. 8A). In addition, there is a significant enrichment of genes downregulated

in luminal B breast cancer (and upregulated in basal breast cancer)[40]. This suggests that PMDs are involved in down-regulation of luminal B-specific genes. Examples of luminal B-downregulated genes include *CD3G*, encoding the gamma polypeptide of the T-cell receptor-CD3 complex (gene sets "signaling" and "adhesion"), and *RBP4*, encoding retinol binding protein 4 (gene set "signaling") (Supplementary Fig. 8B). Stratification of tumors according to low and high median expression of the 400 PMD-downregulated genes revealed significant differences in overall survival of the corresponding patients ($p = 2.6e{-}03$, chi-square test, Supplementary Fig. 8C), suggesting clinical significance of PMD-associated gene repression. Taken together, downregulation of genes inside PMDs occurs rarely and is restricted to low-frequency PMDs. However, these rare cases include genes relevant to breast cancer given the overlap with previously identified luminal B breast cancer-relevant genes and differential overall survival. We finally focused on expression changes of X-linked genes, since the X-chromosome is exceptionally prone to methylation loss (Fig. 1a, Supplementary Fig. 3A). To assess whether this is associated with altered expression of genes involved in the process of X-inactivation (XCI) we regarded *XIST* and genes encoding PRC2 subunits. Multivariate regression revealed that expression of *XIST*, *EED*, and *EZH1/2* is associated with the fraction of chrX inside PMDs ($p = 4.8e{-}05$, Supplementary Fig. 6C, D). To further analyze the effect of PMDs on expression on X-linked genes we stratified X-linked genes according their consensus X-inactivation status (E, escape; S, subject to XCI; VE, variably escaping; PAR, pseudoautosomal region)[41]. Notably, among these categories, escape (E) genes are strongly affected when inside PMDs (Fig. 3l), suggesting a specific sensitivity of escape genes to become repressed when inside PMDs. This was unrelated to altered copy number status of these genes (Supplementary Fig. 6E, see also Supplementary Fig. 1C). Taken together, the fraction of chrX inside PMDs is associated with expression levels of key XCI inactivation genes, and escape genes are specifically sensitive to repression inside X-linked PMDs.

**Reduced DNA methylation in PMDs is a feature of many cancers**. To assess the generality of PMD occurrence in cancer, we extended our analysis to other cancer types and normal tissues. We performed PMD detection in a total of 320 WGBS profiles (133 tumors and 187 normals, from TCGA[35], BLUE-PRINT[36], the Roadmap Epigenomics Project (http://www.roadmapepigenomics.org), and refs. [10,37,38]). Although PMDs are detectable in virtually all tumors and normal tissues (see Methods: data availability), mean DNA methylation inside PMDs is much lower in tumors as compared to normal tissues (Fig. 4a, Supplementary Fig. 9A, $p < 2.2e{-}16$, t-test). PMD methylation levels are not tumor tissue-type specific, as most types display the same range of PMD methylation. However, some tumor tissue types have exceptional low methylation inside PMDs (bladder urothelial carcinoma (BLCA), lung), or lack any loss of methylation (glioblastoma multiforme (GBM), acute lymphoblastic leukemia (ALL), and acute myeloid leukemia (ALL)). Thus, regardless of these extreme cases, absolute levels of PMD methylation do not typify tumor tissue origin, underscoring the variable nature of methylation within PMDs. To assess whether CGI hypermethylation in PMDs is as extensive in these additional tumor types as in breast cancer, we analyzed CGI methylation of these 103 additional tumor samples (Supplementary Fig. 9B, see Methods: data availability). As in breast cancer, extensive hypermethylation of CGIs inside PMDs was consistent in most tumor types, with levels of hypermethylation in Burkitt's lymphoma (BL)[37] being among the highest of all tested tumors

Possibly, these differences are linked to tumor cellularity of the samples. In two GBM and some AML samples, CGI hyper-methylation was not restricted to PMDs, which is suggestive of inaccurate PMD detection due to high methylation inside these tumors' PMDs (see Fig. 4a). Importantly, these results extend the observed tendency of CGI hypermethylation inside PMDs to other tumors.

Lastly, to assess whether the distribution of tumor PMDs reflects tissue of origin we scored the presence of PMDs in genomic tiles of 30 kb and subsequently clustered the resulting binary profiles. The analysis showed that the majority of tumors of the same type clustered together, although not fully accurately (Fig. 4b), suggesting that the genomic distribution of PMDs is linked to tissue of origin. Thus, even though methylation levels of PMDs are mostly independent of tissue-of-origin (Fig. 4a), the distribution of PMDs associates with tissue of origin, likely reflecting differences in the genomic parts that tolerate PMDs.

## Discussion

In this study we analyzed breast cancer DNA methylation profiles to high resolution. The main feature of breast cancer epigenomes is the extensive loss of methylation in PMDs and their hyper-variability. Directly linked to this is the concurrent CGI hyper-methylation, which inside PMDs affects 92% of all CGIs. Although various features of PMDs have been described before, our study is the first to include a larger WGBS cohort from one tumor type, while integrating WGBS data from other tumor types. PMDs may be regarded as tissue-type-specific inactive constituents of the genome: the distribution shows tissue-of-origin specificity, gene expression inside PMDs is low and they are late replicating. Inside PMDs the accumulation of breast cancer mutations is higher than outside of them. The resulting domain-like fluctuation in mutation density is likely related to the fluctuating mutational density along the genome in cancer cells observed by others[42–44]. The phenomena observed in breast cancer extend to tumors of at least 16 additional tissue types underscoring the generality of our findings. We conclude that loss of methylation in PMDs and concurrent CGI hypermethylation is a general hallmark of most tumor types with the exception of AML, ALL, and GBM.

The phenomena that we describe for breast cancer have remained elusive in genome-scale studies that only assessed subsets of the CpGs; the sparsity of included CpGs does not allow accurate PMD detection. Typical analysis strategies include tumor stratification by clustering of the most highly variable CpGs which at least in our breast cancer cohort are located in PMDs. In effect such approaches are biased towards CGIs due to their design and consequently, the hypermethylation groups represent tumors in which PMDs are highly abundant (e.g., see refs. [39,45–53]). It is very likely that for some tumor types hyper-methylation groups associate with clinicopathological features, amongst which a positive association with tumor cellularity is recurrent[46,50–52]. This suggests that PMDs are more pronounced in tumor cells than in the non-tumor tissue of a cancer sample. This makes hypermethylated CGIs useful diagnostic markers but less likely informative as prognostic markers informing about tumor state, progression and outcome.

Since PMDs are domains in which instability at the genetic, epigenetic, and transcriptome level is tolerated, they may provide plasticity that is beneficial for the heterogeneity of tumor cells.

## Methods
**Sample selection, pathology review, and clinical data**. Sample selection, pathology review, and clinical data collection for this study has been described in the ref. [15]. Internal Review Boards of each participating institution approved collection and use of samples of all patients in this study. Samples had previously

been subjected to pathology review and only samples assessed as being composed of >70% tumor cells, were accepted for inclusion in the study. Two independent pathologists assessed paraffin-embedded and frozen sections for all samples, where histological slides were available. Additionally, clinical data was recorded according to the proforma specified by the International Cancer Genome Consortium (ICGC) where possible.

**Processing of whole-genome bisulfite sequencing data**. WGBS library preparation, read mapping, and methylation calling was done as described before[11,54]: genomic DNA (1–2 μg) was spiked with unmethylated λ DNA (5 ng of λ DNA per microgram of genomic DNA; Promega). DNA was shared by sonication to 50–500 bp in size using a Covaris E220 sonicator, and fragments of 150–300 bp were selected using AMPure XP beads (Agencourt Bioscience). Genomic DNA libraries were constructed using the Illumina TruSeq Sample Preparation kit following Illumina's standard protocol. After adapter ligation, DNA was treated with sodium bisulfite using the EpiTect Bisulfite kit (Qiagen), following the manufacturer's instructions for formalin-fixed, paraffin-embedded tissue samples. Two rounds of bisulfite conversion were performed to ensure full conversion. Enrichment for adapter-ligated DNA was carried out through seven PCR cycles using PfuTurboCx Hot-Start DNA polymerase (Stratagene). Library quality was monitored using the Agilent 2100 Bioanalyzer, and the concentration of viable sequencing fragments was estimated using quantitative PCR with the library quantification kit from Kapa Biosystems. Bisulfite converted libraries were paired-end sequenced (2 × 100 nt) on an Illumina Hi-Seq2000. Reads were aligned against the human genome (hg19/GRCh37) using the rmapbs-pe tool from the MethPipe package (v3.0.0)[55] allowing a maximum of 10 mismatches and a maximum fragment length of 600 bp. Adapter sequences were clipped. Mapped reads were sorted according genome position, and duplicates were removed using the duplicate-remover tool from MethPipe (v2.03). Cytosine methylation levels were determined using the methcounts tool from MethPipe (v2.03). All code used for this mapping strategy is made available (see bioinformatic analysis code availability).

**Principal component analysis of WGBS data**. For principal component analysis (PCA) of WGBS profiles, CpGs with coverage of at least 10 were used. Subsequently, the top 5% most variable CpGs were selected. We used the FactoMineR package[20] for R to perform PCA, to determine association of principal components with clinicopathological features, and to perform the corresponding significance testing.

**Detection of PMDs**. Detection of partially methylated domains (PMDs) in all methylation profiles throughout this study was done using the MethylSeekR package for R[56]. Before PMD calling, CpGs overlapping common SNPs (dbSNP build 137) were removed. The alpha distribution[56] was used to determine whether PMDs were present at all, along with visual inspection of WGBS profiles. After PMD calling, the resulting PMDs were further filtered by removing regions overlapping with centromers (undetermined sequence content).

**Mean methylation in PMDs and genomic tiles**. Wherever mean methylation values from WGBS were calculated in regions containing multiple CpGs, the "weighted methylation level"[57] was used. Calculation of mean methylation within PMDs or genomic tiles involved removing all CpGs overlapping with CpG island (-shores) and promoters, as the high CpG densities within these elements yield unbalanced mean methylation values, not representative of PMD methylation. For genome/chromosome-wide visualizations (Fig. 1), 10-kb tiles were used. For visualization, the samples were ordered according hierarchical clustering of the tiled methylation profiles, using "ward.D" linkage and [1-Pearson correlation] as a distance measure.

**Clustering on PMD distribution**. For each sample, the presence of PMDs was binary scored (0 or 1) in genomic tiles of 5 kb. Based on these binary profiles, a distance matrix was calculated using [1-Jaccard] as a distance metric, which was used in hierarchical clustering using complete linkage.

**Tumor suppressor genes and driver mutations**. For overlaps with tumor suppressor genes, Cancer Gene Census (http://cancer.sanger.ac.uk/census, October 2017) genes were used. Overlaps with genes containing breast cancer driver mutations were determined using the list of 93 driver genes as published previously by us[15].

**CIMP**. To determine the association between B-CIMP (fraction of CGIs that are hypermethylated, >30% methylated) and PMD occurrence we used beta-regression using the betareg package in R[58].

**Survival analysis**. Survival analysis of patient groups stratified by expression of genes downregulated in PMDs. For each tumor sample of our breast cancer transcriptome cohort (n = 266[17]), the median expression of all PMD-downregulated genes (Supplementary Data 4) was calculated. The obtained distribution of these medians was used to stratify patient groups, using a two-way split over the median of this distribution. Overall survival analysis using these groups was done using the "survival" package in R, with chi-square significance testing.

**Reporting summary**. Further information on research design is available in the Nature Research Reporting Summary linked to this article.

## Data availability

Tables containing CpG methylation values (bigwig), genomic coordinates and mean methylation values of PMDs and CGIs are available via https://doi.org/10.5281/zenodo.1467025 or https://doi.org/10.17026/dans-276-sda6. Raw data for whole-genome bisulfite sequencing of the 30 breast tumor samples of this study is available from the European Genome-phenome Archive (https://www.ebi.ac.uk/ega) under dataset accession EGAD00001001388 (Study EGAS00001001195, Data Access Committee EGAC00001000010). External data resources used in this study are listed in Supplementary Data 5.

## Code availability

All code for analyses of this study is available on https://github.com/abbrinkman/brcancer_wgbs.git.

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

## Acknowledgements

This work has been funded through the ICGC Breast Cancer Working group by the Breast Cancer Somatic Genetics Study (BASIS), a European research project funded by the European Community's Seventh Framework Program (FP7/2010–2014) under the grant agreement number 242006; the Triple Negative project funded by the Wellcome Trust (grant reference 077012/Z/05/Z). For contributions towards specimens and collections: Tayside Tissue Bank, OSBREAC consortium, Icelandic Center for Research (RANNIS), Swedish Cancer Society, Swedish Research Council, Foundation Jean Dausset-Center d'Etudes du polymorphisme humain, Icelandic Cancer Registry, Brisbane Breast Bank, Breast Cancer Tissue and Data Bank and ECMC (King's College London), NIHR Biomedical Research Center (Guy's and St Thomas's Hospitals), Breakthrough Breast Cancer, Cancer Research UK. We thank E.M. Janssen-Megens, K. Berentsen, H. Kerstens and K.J. Francoijs for technical support. We thank H. Kretzmer and R. Siebert for providing processed data files of the lymphoma dataset. A.B.B. was through the Dutch Cancer Foundation (KWF) grant KUN 2013–5833. SN-Z is personally funded by a CRUK Advanced Clinician Scientist Award (C60100/A23916). M.S. was supported by the EU-FP7-DDR response project. L.B.A. is supported through a J. Robert Oppenheimer Fellowship at Los Alamos National Laboratory. A.L.R. is partially supported by the Dana-Farber/Harvard Cancer Center SPORE in Breast Cancer (NIH/NCI 5 P50 CA168504–02). J.A.F. was funded through an ERC Advanced Grant (ERC-2012-AdG-322737) and ERC Proof-of-Concept Grant (ERC-2017-PoC-767854). A.S. was supported by Cancer Genomics Netherlands (CGC.nl) through a grant from the Netherlands Organization of Scientific research (NWO). We received additional support from the Dutch national e-infrastructure (SURF Foundation). Finally, we would like to acknowledge all members of the ICGC Breast Cancer Working Group.

## Author contributions

A.B.B., S.N-Z., M.R.S., H.S. designed the study, analyzed data and wrote the manuscript. F.S. analyzed data. F.G.R.G. and M.S. contributed towards transcriptomic analyses. A.B. contributed IT processing. S. Martin was the project coordinator. H.D., D.G., M.Ramakrishna, X.Z., J.S., M. Ringnér contributed towards data curation and genomic analyses. L.B.A., S. Morganella contributed towards genomic analyses. A.S., A.F. contributed pathology assessment and/or samples. T.F., V.K. contributed towards DNA methylation data analyses. M.G., I.G.G. contributed to whole-genome bisulfite sequencing experiments/expertize. M.J.v.d.V., A-L.B-D., A.L.R., G.T., J.W.M., J.A.F., drove the consortium and provided samples. All authors discussed the results and commented on the manuscript.
