## [Peer Review File · Nature Communications]

Point-by-point response

Brinkman *et al.*: “Partially methylated domains are hypervariable in breast cancer and fuel widespread CpG island hypermethylation.”

Note: in black, original report text; in blue, our response

Referee expertise:

Referee #1: cancer genomics, epigenetics

Referee #2: cancer epigenetics

Reviewers' Comments:

Reviewer #1:

Remarks to the Author:

In this study, Brinkman et al present an investigation of partially methylated domains (PMDs) in breast cancer primary tumors, comparing PMD features between 30 breast cancer samples that they have profiled by whole genome bisulfite sequencing.

They report a variety of features that are largely previously established for PMDs from prior studies and given the known association of PMDs with LADs and other genome domains. For example, they show that PMDs exhibit a higher mutation frequency, however PMDs are well known to overlap with late replicating regions of the genome, and late replicating regions of the genome are known to exhibit a higher mutation rate (PMID19287383), so this is not unexpected.

The very frequent CGI hypermethylation in PMDs is an interesting observation, which they show is a common feature of tumor DNA methylation. However frequent focal hypermethylation of promoters within PMDs has been identified previously (PMID22120008). Nonetheless, this is investigated in much more detail in this study. Most CGIs within PMDs are methylated to some extent, in contrast to those outside of PMDs.

Analyses undertaken to explore the potential effect of CGI hypermethylation upon gene expression indicate little potential effect, including upon breast cancer TSGs and other breast cancer relevant genes. This is consistent with prior studies in which it has been observed that while promoter hypermethylation in tumors is frequently associated with repressed genes, in normal pre-cancerous cells these genes are often already repressed.

An analysis of PMD occurrence more broadly in a wider set of tumor WGBS datasets showed that they are detectable in essentially all tumors, however this is a well established feature of DNA methylation in cancer, as well as their lower methylation

levels compared to normal tissues. The authors detect PMDs in a range of normal tissues, however the presence of PMDs even more broadly throughout normal tissues was recently reported through analysis of CpGs in the solo-WCGW context (PMID 29610480), an approach that appears to more robustly detect PMDs with greater sensitivity.

The observation that tumor types cluster together based on PMD presence throughout the genome is interesting, potentially indicating that the PMD distribution is linked to the tissue of origin, which may be expected given the significant sharing of PMDs between normal tissues and cancers as reported recently. However, the exploration of this interesting aspect of PMDs is limited in this study.

Overall, this is a clearly written and presented study that provides a deeper investigation of PMD features within a specific type of cancer. However, more sensitive detection of PMDs in multiple samples of a variety of different cancers has been reported recently (PMID 29610480), most of the features associated with PMDs reported here have been published previously, and the PMD features observed in this more extensive set of one cancer type are generally seen more broadly throughout many cancer types, limiting the value of an in-depth analysis of PMDs in many samples of the one cancer type. The primary novel contribution is the more in depth analysis of CGI hypermethylation in PMDs. However, the study does not provide significant advances in addressing the major outstanding questions regarding PMDs, such as the mechanisms driving their formation, the dysfunctional cellular processes in cancer that result in extensive CGI methylation within PMDs, and whether the altered DNA methylation states in PMDs are causal for altered chromatin states or genome activity.

Comments:

1. The authors report that PMDs are hypervariable in size and distribution. Could some PMDs be harder to detect, and could more robust PMD detection using the solo-WCGW signature detect them more reliably?

- We analyzed whether the solo-WCGW method would yield the same or different results regarding this variation.

1. First, we reproduced Figure 1ABC (Suppl. Fig. 3), but exclusively using solo-WCGWs. As reported¹, DNA methylation levels were generally lower (stronger demethylation in solo-WCGWs). However, the observed variation between tumor samples was the same, regardless of whether all CpGs or only solo-WCGWs were used (compare Fig. 1 and Suppl. Fig. 3).

2. Second, we performed PMD calling using only solo-WCGWs and subjected the resulting PMDs to the same analysis as in Fig. 2AB (Suppl. Fig. 4BC). This showed that the genome fraction inside PMDs and the genome fraction in common PMDs was very similar between both PMD callings. Moreover, the union

sets of PMDs resulting from the two PMD calling strategies showed a high degree of overlap (92%, Suppl. Fig. 4D), showing that both strategies do not yield a different set of PMDs.

3. We performed genome segmentation using the solo-WCGW method described above¹ and compared our original PMDs to this aggregate PMD set. This showed that all of sample-specific PMD sets have high overlap with this aggregate PMD track (Suppl. Fig. 4D).

- Taken together, regardless of the different methods used, we detect highly similar PMDs, and the same level of variation.
- It should be noted that the solo-WCGW method of PMD detection¹ is based on cross-sample standard deviation (s.d.) of mean solo-WCGW DNA methylation in 100-kb genomic tiles. The bimodal distribution of these s.d.'s allows for segmentation of the genome into PMD and non-PMD ("HMD") segments, with PMD segments having high s.d.. Thus, this method is strictly dependent on groups of samples and cannot be used to detect PMDs in individual samples. Importantly, as the method uses high cross-sample variation as a classifier for PMDs, it implies that across samples PMDs are highly variable.

2. The extent of PMD variation between tumors is interesting. However, could it be that different tumor samples contain a significantly different mix of cell types, each with a different DNA methylation state, thus leading to variable methylation levels within PMDs based on the fraction of cells in the population that had undergone PMD hypomethylation?

- We do not observe this in this relatively small cohort of 30 samples. We added Suppl. Fig. 1E to show this. However, in the original manuscript we already mentioned in the discussion that such association is being observed in some studies using 450k DNA methylation data. Likely, the number of samples is too limited to detect such association here.

Could this also influence the clustering of tumor types based on PMD presence throughout the genome (Fig. 4C) if the frequency of different cell types is more similar in tumors of the same type than between types?

- This is unlikely, since this clustering is based on location (distribution) of PMDs throughout the genome rather than the level of methylation inside detected PMDs.

3. Fig 3D. As the fraction of methylated CGIs in a genome increases, it would be expected that by chance the subset of them that occur in PMDs would also increase.

- This is not necessarily true, as the fraction of the genome inside PMDs is not

constant but variable between samples. Therefore, we think it is relevant to assess whether this variation can explain the number of hypermethylated CGIs (=CIMP).

4. Fig 3F: Many more tumor WGBS datasets were published recently (PMID 29610480), which could be added to this analysis.

- We added all WGBS data that is currently publicly available. In total, our study now includes data from 320 WGBS profiles in total: 187 normal and 133 tumor samples from 18 different tissue types. To this end we adapted all relevant analyses, and correspondingly changed Fig. 3G, Fig. 4, and Suppl. Fig. 9 (was Suppl. Fig. 7).

Reviewer #2:

Remarks to the Author:

Partially methylated domains are hypervariable in breast cancer and fuel widespread CpG island hypermethylation

Brinkman et al.

The authors describe a comprehensive partially methylated domains (PMDs) characterization using whole genome bisulfite sequencing (WGBS) from a cohort of 30 breast tumor samples. The authors find that previously described loss of methylation in cancer PMDs is also confirmed in their cohort. This loss of methylation takes a large genome fraction and it is very variable in terms of extension and distribution, representing the most important source of epigenetic variation in the cohort. Moreover, it is not generally related neither to genomic gains nor aberrant expression of genes involved in 5-methylcytosine modification. The authors also find that PMDs are probably repressive domains, as they harbor some repressive signals such as LaminB1 or local increase in CTCF binding at the borders. Accordingly, the authors show that PMDs have low gene density and expression levels. In addition, an increase in genomic instability is also observed. Stratification analysis evidence a general intermediate methylated state, including CpG islands, which are generally hypomethylated outside PMDs. The authors also characterize the relation between PMD demethylation and gene expression, finding that widespread cancer-associated gene repression inside PMDs is limited. Finally, the authors find that, whilst the presence of PMDs is not exclusive of cancer tissues, tumor PMDs present a reduced DNA methylation and their distribution is related to tissue of origin.

Global loss of DNA methylation and CpG island (CGI) hypermethylation are well-recognized hallmarks of cancer. The most interesting finding of this study is the stronger effect in CpG methylation in CGIs located PMDs. Most of the CGIs inside PDMS lose their strictly hypomethylated state and become more methylated, in

comparison with CGIs outside PMDs, where only around 25% shown hypermethylation (Fig. 3B). Importantly, number of CGIs inside and outside of PMDs are similar: ~13.000 vs. ~15.000. However, at the moment, this fact is not significantly associated to biological consequences, as no obvious changes in gene expression have been identified, reducing considerably the impact of the finding.

Comments:

1. Regarding samples included in the study, 25 ER-positive and 5 ER-negative, ER status should be included in Fig. 1A (as shown in Suppl. Fig. 2A).

Overall, there are at least one ER- sample harboring a profound hypomethylation (PD10014a). In addition, data depicted in Fig. 1D shown that the second-largest source of variation is actually ER status. Therefore, the statement "...that lacked obvious association with ER-status" should be moderated, as only 5 ER- samples have been included in the analysis.

- We adapted Fig. 1A accordingly (as we did for the newly included Fig. 4B), to indicate ER status.
- Even though at least one ER- sample with profound hypomethylation is present, there is no consistent trend among other ER- samples. Nevertheless, we deleted this statement to avoid any confusion. In our initial manuscript text, we already noted that our sample size (n=30) is relatively small for detection of weaker associations.

Moreover, additional analyses considering other clinicopathological data, including subtypes (basal, luminal A, luminal B) could be included, more considering the results from GEA ("...significant enrichment of genes downregulated in luminal B breast cancer").

- According the reviewer's comments we have included Suppl. Fig. 4A. In addition, we included text stating that these 'intrinic' subtypes (AIMS) are associated with principal component 2, (PC2) but that this is likely confounded with ER status, which is also associated with PC2.

2. The "exceptionally prone methylation loss" detected in chromosome X, should be discussed considering the X inactivation event.

- We have included text and figures (Fig. 3L, Suppl. Fig. 6CDE) presenting analyses that address this point. We found that expression levels of XIST, EED, and EZH1/2 (i.e. genes involved in the process of X-inactivation) are specifically associated with the fraction PMDs within chrX. Furthermore, we found that genes normally escaping from X-inactivation are downregulated when inside PMDs, suggesting that these genes are specifically sensitive to PMD effects.

3. Although 72 WGBS from normal tissues are analyzed, data from normal breast tissue are not included in the analysis, and considering the study, at least a small set should be included.

- Normal breast samples were unfortunately not included in the BASIS cohort, for none of the analyses platforms used by the BASIS Working Group (WGS, RNA-seq²⁻⁴). We sought to use publicly available WGBS data for normal breast. During our analyses resulting in the revision, only two normal breast samples were publicly available from the Roadmap Epigenomics Project (GEO: GSM613869, GSM613869), but this data unfortunately lacks coverage information, which makes PMD detection impossible. The raw data for these samples has not been released for (controlled) access yet.

4. Although the results about the effect of PMDs in methylation are convincing, statements as “breast tumor whole-genome DNA methylation profiles reveal global loss of methylation due to PMDs”, or “PMDs appear to be a major driver or even causal” can not be established with the data provided from the study, as causality has not been proved (only association).

- We have changed these sentences to
 1. “breast tumor whole-genome DNA methylation profiles reveal global loss of methylation in structures known as PMDs” (section “Primary breast tumors display variable loss of DNA methylation”).
 2. “Directly linked to this is the concurrent CGI hypermethylation, which inside PMDs affects 92% of all CGIs” (Discussion).

5. Regarding the cohort of samples used to define the effect in gene expression, considering PMD frequencies have been derived from 30 samples, the subset of 24 overlapping cases with WGBS and transcriptomic data should be used in main figures. Thus, data shown in Fig. 2 should be replaced by data shown in Suppl. Fig. 3. Data in Fig. 2 (n=266) can be included as supplementary information. Data about gene coding density, now in Suppl. Fig. 3A could be shown in main figure 2.

- We applied all changes as suggested, in both figures as well as in the manuscript text.
- Please note that we encountered an error that we corrected throughout the manuscript: 25 (not 24) of our WGBS samples overlapped with the transcriptome dataset⁴.

6. Could you please include the figure for this data “In our cohort of 560 full breast cancer genomes, substitutions, insertions, and deletions occur more frequently within than outside PMDs”

- This was already present in the original manuscript (Fig. 2G). After reorganizing the figure order (previous comment) this information is now present in Suppl. Fig. 5B

7. Could you please include a figure showing the distribution of PMDs by gene location (% of PMDs occurring in promoters, CGIs, gene body, enhancers...) to complement Fig. 2H?

- As the size of PMDs is much larger than any of the genomic elements mentioned by the reviewer, such figure would be very non-informative (e.g. an average PMD -- up to Mb in size -- cannot be located inside a promoter). We therefore did not include such figure.

8. Suppl. Fig. 4C, showing the number of CGIs inside and outside of PDMs, must be included in main Figure 3.

- Accordingly, we moved this figure to the main figures (now Fig. 3D).

9. In Fig. 4B, it could be more accurate to label “primary tissue” as “normal tissue”, the same in “cultured primary cells”, to emphasize the fact that they are normal samples.

- As we have now included a total of 320 WGBS samples in our analysis, the layout of this figure needed to change to accommodate this, and the labeling of samples is adapted according the reviewer’s suggestion.

10. In Fig. 3A, are the findings occurring in the samples with PMDs? “When individual PMDs are regarded, CGIs inside of them lose their strictly hypomethylated state and become more methylated to a degree that varies between tumors (Fig. 3A)”.

- The phrase cited by the reviewer likely relates to the region indicated with red bars, which indicate PMDs detected for each of 30 samples. The lower panels of Fig. 3A includes data from all 30 samples, also those that have no PMDs at this location, as indicated at the y-axis label (‘tumors’ and ‘1 to 30’)

11. Regarding the effect of PMDs over expression of TSGs, it could be worthy to limit the analysis to TSGs with CGIs inside PMDs. Did you try it?

- The analysis suggested by the reviewer is essentially what is presented in Suppl. Fig. 7C (Suppl. Fig. 5C of the original manuscript). Here we show the Pearson correlation between CGI-methylation and expression for tumor suppressor genes (red) and breast cancer driver-mutated genes (brown), while also including that of other genes (green).

References

1. Zhou, W. *et al.* DNA methylation loss in late-replicating domains is linked to mitotic cell division. *Nature Genetics* **50**, 591–602 (2018).
2. Nik-Zainal, S. *et al.* Landscape of somatic mutations in 560 breast cancer whole-genome sequences. *Nature* **534**, 1–20 (2016).
3. Morganella, S. *et al.* The topography of mutational processes in breast cancer genomes. *Nature Communications* **7**, 11383 (2016).
4. Smid, M. *et al.* Breast cancer genome and transcriptome integration implicates specific mutational signatures with immune cell infiltration. *Nature Communications* **7**, 12910 (2016).

REVIEWERS' COMMENTS:

Reviewer #2 (Remarks to the Author):

The authors have addressed my previous comments.

Reviewer #3 (Remarks to the Author):

In this revised version of the manuscript by Brinkman et al., the authors satisfactorily addressed all points raised previously by reviewer 1. Notably, they repeated the analyses to detect and characterize PMDs using the solo-WCGW method, and convincingly show that the results obtained with their method and the solo-WCGW method are very similar. In general, as commented by the authors, a method for detection of PMDs independent of the analyzed group of samples seems advantageous.

Further comments:

The authors report that PMD methylation is the main source of variation in the breast cancer methylome profiles (Figure 1D), without providing an explanation for this observation. A previous study (PMID29610480) has linked methylation loss in PMDs to mitotic cell division. Can this be confirmed in the present study, e.g. by comparison with an epigenetic mitotic clock or an expression-based mitotic index (PMID27716309)?

The authors further report that PMD methylation variability is not associated to clinicopathological parameters except ER status and as a consequence, molecular breast cancer subtypes. This is partly due to the small study size, which limits detectability to the strongest associations. Since only 25/30 analyzed samples are contained in Ref. 15, a table covering clinical parameters for all 30 patients included in this study will enhance transparency and should be informative for subsequent analyses.

** See Nature Research's author and referees' website at www.nature.com/authors for information about policies, services and author benefits

Please find below in response to the reviewers' comments bullet-pointed in blue.

REVIEWERS' COMMENTS:

Reviewer #2 (Remarks to the Author):

The authors have addressed my previous comments.

- We have no comments and thank the reviewer

Reviewer #3 (Remarks to the Author):

In this revised version of the manuscript by Brinkman et al., the authors satisfactorily addressed all points raised previously by reviewer 1. Notably, they repeated the analyses to detect and characterize PMDs using the solo-WCGW method, and convincingly show that the results obtained with their method and the solo-WCGW method are very similar. In general, as commented by the authors, a method for detection of PMDs independent of the analyzed group of samples seems advantageous.

Further comments:

The authors report that PMD methylation is the main source of variation in the breast cancer methylome profiles (Figure 1D), without providing an explanation for this observation. A previous study (PMID29610480) has linked methylation loss in PMDs to mitotic cell division. Can this be confirmed in the present study, e.g. by comparison with an epigenetic mitotic clock or an expression-based mitotic index (PMID27716309)?

- We have analyzed this in more detail, using the 'mitotic score' of the tumor samples used in our study. The result is shown in the figure below. In addition to mean PMD methylation (left-most panel), we found mitotic score to be the only significantly associated clinical feature with principal component PC1 ($p=1.7e-04$). The observed association (higher mitotic score with higher PC1 score) is similar to that of PMD methylation. This is the opposite from what is reported in the two studies mentioned by the reviewer (lower PMD methylation linked to higher mitotic cell division). In addition, the observed trend is not consistent over the full range of PC1 scores, as observed for mean PMD methylation. Thus, our results do not confirm the observations of the mentioned studies. We currently cannot explain this discrepancy but are careful as the sample size ($n=30$ tumors) of our study may be rather limited to draw firm conclusions. We chose to not include the results of this analysis in the manuscript.

Association of variation (PC1, PC2) with mitotic score

The authors further report that PMD methylation variability is not associated to clinicopathological parameters except ER status and as a consequence, molecular breast cancer subtypes. This is partly due to the small study size, which limits detectability to the strongest associations. Since only 25/30 analyzed samples are contained in Ref. 15, a table covering clinical parameters for all 30 patients included in this study will enhance transparency and should be informative for subsequent analyses.

- We have included the clinical parameters for the used tumor samples in a new Supplementary Table (Supplementary Table 2 of the revised manuscript).